# Elevated Fasting Blood Glucose Levels Are Associated with Worse Clinical Outcomes in COVID-19 Patients Than in Pneumonia Patients with Bacterial Infections

**DOI:** 10.3390/pathogens11080902

**Published:** 2022-08-10

**Authors:** Wenjun Wang, Zhonglin Chai, Mark E Cooper, Paul Z Zimmet, Hua Guo, Junyu Ding, Feifei Yang, Xixiang Lin, Xu Chen, Xiao Wang, Qin Zhong, Zongren Li, Peifang Zhang, Zhenzhou Wu, Xizhou Guan, Lei Zhang, Kunlun He

**Affiliations:** 1Key Laboratory of Ministry of Industry and Information Technology of Biomedical Engineering and Translational Medicine, Chinese PLA, General Hospital, Beijing 100853, China; 2Beijing Key Laboratory for Precision Medicine of Chronic Heart Failure, Chinese PLA, General Hospital, Beijing 100853, China; 3Medical Big Data Research Center, Chinese PLA, General Hospital, Beijing 100853, China; 4Central Clinical School, Faculty of Medicine, Monash University, Melbourne 3004, Australia; 5Department of Pulmonary and Critical Care Medicine, Chinese PLA, General Hospital, Beijing 100853, China; 6BioMind Technology, Zhongguancun Medical Engineering Center, 10 Anxiang Road, 8th Floor, Beijing 100872, China; 7China-Australia Joint Research Center for Infectious Diseases, School of Public Health, Xi’an Jiaotong University Health Science Center, Xi’an 710061, China; 8Artificial Intelligence and Modelling in Epidemiology Program, Melbourne Sexual Health Centre, Alfred Health, Melbourne 3004, Australia; 9Department of Epidemiology and Biostatistics, College of Public Health, Zhengzhou University, Zhengzhou 450001, China

**Keywords:** COVID-19, pneumonia patients, fasting blood glucose, clinical severities

## Abstract

Aims: We investigate how fasting blood glucose (FBG) levels affect the clinical severity in coronavirus disease 2019 (COVID-19) patients, pneumonia patients with sole bacterial infection, and pneumonia patients with concurrent bacterial and fungal infections. Methods: We enrolled 2761 COVID-19 patients, 1686 pneumonia patients with bacterial infections, and 2035 pneumonia patients with concurrent infections. We used multivariate logistic regression analysis to assess the associations between FBG levels and clinical severity. Results: FBG levels in COVID-19 patients were significantly higher than in other pneumonia patients during hospitalisation and at discharge (all *p* < 0.05). Among COVID-19 patients, the odds ratios of acute respiratory distress syndrome (ARDS), respiratory failure (RF), acute hepatitis/liver failure (AH/LF), length of stay, and intensive care unit (ICU) admission were 12.80 (95% CI, 4.80–37.96), 5.72 (2.95–11.06), 2.60 (1.20–5.32), 1.42 (1.26–1.59), and 5.16 (3.26–8.17) times higher in the FBG ≥7.0 mmol/L group than in FBG < 6.1 mmol/L group, respectively. The odds ratios of RF, AH/LF, length of stay, and ICU admission were increased to a lesser extent in pneumonia patients with sole bacterial infection (3.70 [2.21–6.29]; 1.56 [1.17–2.07]; 0.98 [0.88–1.11]; 2.06 [1.26–3.36], respectively). The odds ratios of ARDS, RF, AH/LF, length of stay, and ICU admission were increased to a lesser extent in pneumonia patients with concurrent infections (3.04 [0.36–6.41]; 2.31 [1.76–3.05]; 1.21 [0.97–1.52]; 1.02 [0.93–1.13]; 1.72 [1.19–2.50], respectively). Among COVID-19 patients, the incidence rate of ICU admission on day 21 in the FBG ≥ 7.0 mmol/L group was six times higher than in the FBG < 6.1 mmol/L group (12.30% vs. 2.21%, *p* < 0.001). Among other pneumonia patients, the incidence rate of ICU admission on day 21 was only two times higher. Conclusions: Elevated FBG levels at admission predict subsequent clinical severity in all pneumonia patients regardless of the underlying pathogens, but COVID-19 patients are more sensitive to FBG levels, and suffer more severe clinical complications than other pneumonia patients.

## 1. Introduction

Pneumonia is a severe and life-threatening condition caused by viral, bacterial, and fungal infections [1,2]. Prior to the COVID-19 pandemic, the most common clinically reported pneumonia cases were caused by bacterial infections, which often lead to severe clinical adversities [3,4]. However, bacterial and fungal pathogens are not highly infectious, and usually only affect certain vulnerable individuals in clinical settings, such as diabetes and AIDS patients. In 2002 and 2012, the pneumonia outbreaks caused by severe acute respiratory syndrome coronavirus-1 and Middle East respiratory syndrome coronavirus, respectively, developed into sizable epidemics [5]. They were eventually eliminated or suppressed at a low endemic level with stringent public health measures. In contrast, COVID-19 has caused an unprecedented pandemic since 2019, and remains a global health emergency [6]. By 17 May 2021, more than 162.7 million confirmed cases of COVID-19 were recorded worldwide, including more than 3.3 million deaths [7].

Similar to pneumonia caused by other pathogens, severe acute respiratory syndrome coronavirus-2 (SARS-CoV-2) is capable of causing severe systemic inflammatory responses and multiple organ failure, ultimately leading to death [8]. In addition to respiratory complications, SARS-CoV-2 may cause severe damage to the cardiovascular system, with complications including myocarditis, acute myocardial injury, and heart failure, as well as acute liver injury and acute kidney injury. A large body of scientific evidence has shown that significant injury to the cardiovascular system, liver, and kidneys indicates a poor prognosis in SARS-CoV-2-infected individuals. Compared with other pathogens, SARS-CoV-2 is far more contagious and has a very high mutation rate, making it very difficult to prevent and treat. While antibiotics are effective in treating bacterial pneumonia, currently, there is no highly effective antiviral treatment for SARS-CoV-2 infection [9].

Apart from the complications mentioned above, previous studies have also shown that increased C-reactive protein levels, diabetes, hyperglycaemia, and other clinical indicators are associated with an increased risk of mortality in pneumonia patients [3,8], among which diabetes and hyperglycaemia are strong risk factors for mortality in community-acquired bacterial pneumonia (CAP-bacteria) [10]. Recent clinical studies have shown a similar and even stronger association between diabetes or hyperglycaemia and COVID-19 [11]. Patients with diabetes or hyperglycaemia are reportedly at a much higher risk of poor prognoses in COVID-19 [11,12]. A report showed that more than 20% of hospitalised COVID-19 patients had diabetes [12]. The Austrian study demonstrated that 23–25% of COVID-19 patients with diabetes had a poor prognosis [12]. COVID-19 patients with elevated FBG levels were 1.58 and 3.22 times more likely to progress to severe/critical stages and death, respectively [13]. Recent evidence further suggests that the SARS-CoV-2 infection may trigger new-onset diabetes in infected patients [14]. SARS-CoV-2 binds to angiotensin-converting enzyme 2 (ACE2), which is found to be expressed in various tissues, including in pancreatic cells, and leads to abnormal glucose metabolism, which complicates the pathophysiology of pre-existing diabetes [14,15]. Inflammation caused by SARS-CoV-2 infection can impair peripheral glucose uptake that alters membrane permeability [11,16,17]. An estimated 463 million individuals worldwide had diabetes in 2019, and about 1.6 million deaths are directly attributed to diabetes each year [17,18]. The two large and expanding epidemics of COVID-19 and diabetes have now collided at a global level, resulting in an unprecedented public health emergency that may profoundly impact human health now and for decades to come.

Despite the mounting evidence indicating that they are risk factors for increasing COVID-19 severity and mortality [10,19,20,21,22,23,24], it remains unknown whether diabetes or increased FBG levels would have the same or a significantly greater impact on the clinical severity of COVID-19 compared to pneumonia patients with bacterial and fungal infections. Therefore, a study on pneumonia patients with infection by various pathogens in a similar setting would be valuable to shed light on the underlying differences. By comparing COVID-19 and pneumonia patients with bacterial infections or concurrent bacterial and fungal infections, we aimed to investigate how FBG levels impact the prognosis of pneumonia patients caused by different pathogens in a similar clinical setting.

## 2. Methods

### 2.1. Participants, Inclusion and Exclusion Criteria, and Clinical Discharge Criteria

We established a retrospective observational cohort study based on 3059 COVID-19 patients admitted to Huoshenshan Hospital in Wuhan—an emergency hospital rapidly constructed and dedicated to COVID-19 patients—from 4 February to 15 April 2020, along with 4615 pneumonia patients admitted to the Chinese PLA General Hospital in Beijing from 6 April 2012 to 24 April 2020. The majority of the pneumonia patients were from the pre-COVID-19 era, and those admitted to the hospital after COVID-19 emerged were confirmed to be non-COVID-19 patients, whereas the COVID-19 patients were transferred to other specific hospitals for treating COVID-19 patients. The two hospitals and their laboratories are under the same administration; thus, their workers, laboratory tests, and treatments are similar. We categorised the participants into three groups: COVID-19 patients, pneumonia patients with sole bacterial infection, and those with concurrent bacterial and fungal infections. Details of inclusion and exclusion criteria and hospital discharge criteria are provided in Appendix A.

### 2.2. Clinical and Outcome Indicators, and Study Strata

We collected data including all of the clinical and outcome indicators of the participants from the hospitals’ electronic medical records. Demographic data, signs and symptoms, and information on the subjects’ pre-existing comorbidities (i.e., coronary heart disease (CHD), cancer, chronic bronchitis (CB), cerebrovascular disease, chronic kidney disease (CKD), chronic obstructive pulmonary disease (COPD), diabetes, hepatitis, and hypertension) were recorded within 24 h of hospital admission. The laboratory data were determined within 72 h of hospital admission. Pre-existing comorbidities were diagnosed upon admission based on the patients’ medical history and point-of-care diagnosis according to the ICD10-CM code [25].

We defined certain events—including clinical complications, ICU admission, and the length of stay—as key clinical outcomes in this study. Clinical complications included acute respiratory distress syndrome (ARDS), acute myocardial injury and heart failure (AMI/HF), acute hepatitis/liver failure (AH/LF), respiratory failure (RF), shock, and acute kidney injury (AKI). According to the Chinese guidelines for the prevention and control of type 2 diabetes (2017 edition) [26], we used the FBG level as the key parameter to stratify the groups for analysis. Detailed definitions for clinical symptoms, complications, and FBG groups are provided in the Appendix A.

### 2.3. Statistical Analysis

We presented continuous variables as the median and interquartile range (IQR), and examined the differences between disease severity groups using the Kruskal–Wallis one-way ANOVA. We presented categorical variables with corresponding percentages, and examined the differences using the χ2 test or Fisher’s exact test. We collected the data on FBG levels at three time points: FBG levels at admission, FBG levels during hospitalisation, and FBG levels at discharge. The FBG levels were presented as the median and IQR. The Kruskal–Wallis one-way ANOVA and the Nemenyi test were used to examine the differences among the three pneumonia groups and across the three time points. The multivariate logistic regression and linear regression analysis models—adjusted for age, sex, and pre-existing comorbidities (except for diabetes)—were used to assess the associations between FBG levels and clinical outcomes. The interaction was used to assess the impact of interactions among FBG levels and the three pneumonia groups on clinical outcomes, as previously described [27]. Data imputation was performed for indicators with a missing percentage < 30% using multivariate imputation by chained equations. A *p*-value < 0.05 was considered statistically significant. Statistical analyses were conducted using R software (version 3.6.1, R Core Team, Vienna, Austria).

## 3. Results

### 3.1. Demographic and Baseline Characteristics of Patients

We included 2761 COVID-19 patients, 1686 pneumonia patients with sole bacterial infection, and 2035 pneumonia patients with concurrent infections (Figure 1). Table 1 shows the basic demographic characteristics and pre-existing comorbidities of these pneumonia patients. Pneumonia patients with concurrent infections were consistently older than patients with sole bacterial infection (median 65 [52–79] years versus 59 [47–69] years) and COVID-19 patients (median 65 [52–79] years versus 60 [50–68] years). There was a greater male bias in pneumonia patients with sole bacterial infection than in patients with concurrent infections or COVID-19 patients (69.48% vs. 68.67% and 50.74%, respectively). In particular, COVID-19 patients demonstrated significantly lower rates of pre-existing comorbidities than pneumonia patients with sole bacterial infection or concurrent infections. These included less CHD, cancer, CB, cerebrovascular disease, CKD, diabetes, hepatitis, and hypertension (all *p* < 0.05).

Table 1 shows the signs and symptoms, laboratory findings, and clinical outcomes of the participants. More COVID-19 patients presented common signs and symptoms than pneumonia patients with sole bacterial infection or concurrent infections, including cough, shortness of breath, diarrhoea, and fatigue. However, COVID-19 patients had a significantly lower incidence of inflammation and multiple organ failure, as reflected by lower plasma levels of C-reactive protein, D-dimer, white blood cell count, lactate dehydrogenase, thrombinogen time, serum albumin, fibrinogen, creatinine, cystatin C, and creatine kinase. Similarly, significantly fewer COVID-19 patients had serious complications, including AMI/HF (1.63% vs. 29.30% and 40.98%, respectively), AH/LF (1.59% vs. 22.89% and 32.19%, respectively), RF (1.67% vs. 5.10% and 16.27%, respectively), shock (0.54% vs. 0.65% and 3.00%, respectively), or AKI (0.29% vs. 23.78% and 30.57%, respectively). However, significantly more COVID-19 patients developed ARDS (0.76% vs. 0.18% and 0.25%, respectively). In terms of clinical outcomes, COVID-19 patients and pneumonia patients with bacterial infection had a significantly shorter length of stay than those with concurrent infections (14.00 and 14.00 vs. 20.00 days, respectively). COVID-19 patients had a significantly lower percentage of ICU admission than the other groups (3.91% vs. 5.34% and 7.81%, respectively) (all *p* < 0.05).

### 3.2. Comparison of FBG Levels among Pneumonia Groups

Figure 2 shows the different trends of changes in FBG levels among the three pneumonia groups after admission. Among the patients with FBG < 6.1 mmol/L at admission, their median levels of FBG were different among these three pneumonia groups at admission, with the lowest median FBG levels in COVID-19 patients and the highest median FBG levels in the pneumonia patients with concurrent infections. The difference remained during the hospitalisation and at discharge among these three groups (Kruskal–Wallis one-way ANOVA; all *p* < 0.05). Further pairwise comparison among these groups showed that FBG levels in pneumonia patients with concurrent infections were significantly higher than in other groups (Nemenyi; all *p* < 0.05) at all three time points (Figure 2, left panel).

Among patients with either FBG 6.1–6.9 mmol/L (Figure 2, middle panel) or FBG ≥ 7.0 mmol/L (Figure 2, right panel) at admission, there were no differences in their median FBG levels at admission among the three groups. With the same basal levels of FBG at admission, their median FBG levels became significantly different among the three groups, both during hospitalisation and at discharge (Kruskal–Wallis one-way ANOVA; all *p* < 0.05). Pairwise comparison showed that the FBG levels in COVID-19 patients were significantly higher than in the other groups at discharge (Nemenyi; all *p* < 0.05) (Figure 2).

### 3.3. The Odds Ratios of Complications and Clinical Outcomes Stratified by Pneumonia Groups

Figure 3 shows the adjusted odds ratios of complications and clinical outcomes of patients with FBG > 7 mmol/L and 6.1–6.9 mmol/L at admission compared to those with FBG < 6.1 mmol/L at admission, stratified by pneumonia group. Among COVID-19 patients, the FBG 6.1–6.9 mmol/L group showed no significant difference in the odds ratios of complications, but the length of stay and the ICU admission were both significantly higher (OR = 1.18, 1.01–1.37, *p* = 0.035; 2.45, 1.22–4.61, *p* < 0.001). In contrast, most of the odds ratios of complications and clinical outcomes showed no significant differences across the FBG level groups in pneumonia patients with sole bacterial infection or concurrent infections. In both patient groups—sole bacterial infection and concurrent infections—only AH/LF (1.47, 1.01–2.12, *p* = 0.040; 1.32, 1.01–1.73, *p* = 0.047) and RF (2.57, 1.28–4.95, *p* = 0.006; 1.67, 1.17–2.36, *p* = 0.004) were significantly higher in the FBG 6.1–6.9 mmol/L group compared with the FBG < 6.1 mmol/L group. Among pneumonia patients with concurrent infections, the odds ratio of ICU admission (1.62, 1.01–2.58, *p* = 0.043) was significantly higher in the group with FBG 6.1–6.9 mmol/L when compared with the FBG < 6.1 mmol/L group.

COVID-19 patients with FBG ≥ 7.0 mmol/L at admission showed significantly higher odds of ARDS and a longer hospital stay (12.80, 4.80–37.96, *p* < 0.001; 1.42, 1.26–1.59, *p* < 0.001, respectively) than COVID-19 patients with FBG < 6.1 mmol/L at admission. Both ARDS and length of hospital stay showed no significant differences in pneumonia patients with FBG ≥ 7.0 mmol/L vs. FBG < 6.1 mmol/L at admission with sole bacterial infection or concurrent infections. In all three patients groups, the odds ratios of AH/LF (2.60, 1.20–5.32, *p* = 0.011; 1.56, 1.17–2.07, *p* = 0.002; 1.21, 0.97–1.52, *p* = 0.085), RF (5.72, 2.95–11.06, *p* < 0.001; 3.70, 2.21–6.29, *p* < 0.001; 2.31, 1.76–3.05, *p* < 0.001), and ICU admission (5.16, 3.26–8.17, *p* < 0.001; 2.06, 1.26–3.36, *p* = 0.004; 1.72, 1.19–2.50, *p* = 0.004, respectively) were significantly higher in the patients with FBG ≥ 7.0 mmol/L than those with FBG < 6.1 mmol/L at admission. Furthermore, pneumonia patients with concurrent infections showed a significantly higher odds ratio of shock (2.41, 1.34–4.46, *p* = 0.004) when they had FBG ≥ 7.0 mmol/L in comparison with those with FBG < 6.1 mmol/L at admission.

Appendix A) shows the interactions between complications and clinical outcomes of patients, their FBG levels at admission, and their pneumonia aetiologies. Among patients with FBG ≥ 7.0 mmol/L, COVID-19 patients showed significantly higher odds of AH/LF, RF, ICU admission, and a longer hospital stay (2.60 vs. 1.56, *p* = 0.015; 5.72 vs. 3.70, *p* = 0.009; 5.16 vs. 2.06, *p* = 0.001; 1.42 vs. 0.98, *p* = 0.021) than pneumonia patients with sole bacterial infection. They also exhibited significantly higher odds of ARDS, AH/LF, RF, ICU admission, and a longer hospital stay (12.80 vs. 3.04, *p* = 0.029; 2.60 vs. 1.21, *p* = 0.029; 5.72 vs. 2.31, *p* = 0.002; 5.16 vs. 1.72, *p* < 0.001; 1.42 vs. 1.02, *p* = 0.033) than pneumonia patients with concurrent infections.

### 3.4. FBG Levels at Admission Predict the Risk of ICU Admission during Hospitalisation

Figure 4 shows the cumulative incidence curves of ICU admission during hospitalisation in all pneumonia groups, stratified by their FBG levels at admission. A high FBG level at admission predicted a higher incidence of ICU admission in all pneumonia groups. Among COVID-19 patients, on day 21 of hospitalisation, the incidence rate was 12.30% [8.61–15.90%] in patients with FBG ≥ 7.0 mmol/L at admission—significantly higher than that of patients with FBG 6.1–6.9 mmol/L (7.24% [3.12–11.36%]) or FBG < 6.1 mmol/L at admission (2.21% [1.55–2.87%] (*p* < 0.001). Similarly, ICU admission rates on day 21 during hospitalisation of pneumonia patients with a sole bacterial infection were 7.49% [5.00–9.99%], 6.91% [3.29–10.54%] and 3.65% [2.49–4.80%] (*p* = 0.003) for patients with FBG levels at admission of ≥7.0, 6.1–6.9, and <6.1 mmol/L, respectively. Pneumonia patients with concurrent infections showed a similar pattern of ICU admission rate, with rates of 8.99% [6.87–11.10%], 7.95% [4.90–11.00%] and 4.37% [3.09–5.64%] (*p* = 0.002) for those with FBG levels at admission of ≥7.0, 6.1–6.9, and <6.1 mmol/L, respectively.

## 4. Discussion

Our study evaluated the influence of FBG levels on the risk and severity of complications in COVID-19 patients, pneumonia patients with sole bacterial infection, and patients with concurrent infections. We report that an elevated FBG level at admission predicts a higher subsequent risk and severity of complications—particularly the risk of ICU admission—regardless of the underlying pathogens among the pneumonia patients. However, the adverse impacts are greater in COVID-19 patients than in other pneumonia groups, with the odds ratios of ARDS, RF, AH/LF, length of hospital stay, and ICU admission being 13, 6, 3, 2, and 5 times higher, respectively, in individuals with FBG ≥ 7.0 mmol/L compared to those with FBG < 6.1 mmol/L at admission. The corresponding odds ratios are also higher in the other pneumonia groups, but in general are 2–4 times lower when compared to COVID-19 patients. Furthermore, COVID-19 patients with a higher FBG level at admission tend to sustain a much higher FBG level throughout hospitalisation and at discharge than the other pneumonia groups.

Our study reveals that an elevated FBG level at the time of hospital admission generally indicates a worse prognosis among pneumonia patients, regardless of the underlying aetiology. This finding is consistent with previous findings. Jensen et al., in 2016, demonstrated that in a cohort of 1318 patients with CAP, an increased FBG was associated with a prolonged length of hospital stay, an increased risk of ICU admission, and increased mortality [28]. In 2020, Wang et al. demonstrated in a multicentre retrospective study of COVID-19 that FBG ≥ 7.0 mmol/L is a risk factor for ICU admission, in-hospital mortality, and complications [29]. Patients with diabetes or hyperglycaemia generally demonstrate defects in innate and adaptive immunity, substantially reducing the host immune response to infection, thereby reducing their ability to eliminate pathogens, and compromising other immune functions needed to fight against the invading pathogens [30,31,32]. These immune defects subsequently lead to a faster disease progression to more severe clinical complications [33].

Our study further indicates that COVID-19 patients may experience much higher risks of complications during hospitalisation if their FBG levels are ≥7.0 mmol/L at admission, when compared with other pneumonia groups. For community-acquired pneumonia caused by bacteria, the host immune response is regulated by both pro-inflammatory and anti-inflammatory cytokines. The infection first triggers the activation of alveolar macrophages, leading to the secretion of pro-inflammatory cytokines and chemokines. A pro-inflammatory amplification loop is established among recruited macrophages, lymphocytes, and polymorphonuclear neutrophils. This mechanism is often counterbalanced by anti-inflammatory cytokines that confine the inflammatory response in the microenvironment and downregulate these responses [34]. In contrast, SARS-CoV-2 infection often triggers a strong inflammatory response, leading to multiple organ damage [35,36]—partially as a result of a hyperinflammatory syndrome and an inflammatory cytokine storm [37]. Notably, pro-inflammatory molecules that influence relevant pathways promoting an inflammatory cytokine storm in SARS-CoV-2 infection could arguably also be involved in the pathophysiology of diabetes [38]. Immune system dysregulation as a result of SARS-CoV-2 infection mainly acts on an inflammatory pathway involving NF-κB—a nuclear factor enhancer of activated B cells that acts as a master switch to regulate pro-inflammatory gene transcription, leading to increased cytokine production and damage to affected organs. Diabetes is an inflammatory condition where sustained high NF-κB activity is observed. We consider that SARS-CoV-2 infection and diabetes can affect common inflammatory pathways [39]. Hence, the interaction of COVID-19 and diabetes promotes a vicious cycle of cytokine release and an impaired glucose metabolism, resulting in systemic inflammation and various severe clinical complications [35]. The lack of effective antiviral agents for SARS-CoV-2 infection also means that it remains more difficult to downregulate the immune response and control disease progression clinically [40], compared with treatments for bacterial- and fungal-induced pneumonia [41].

Our study indicates that the FBG levels remain high throughout hospitalisation among COVID-19 patients admitted with initially elevated FBG levels. This contrasts with the substantially reduced FBG levels during hospitalisation and at discharge among other pneumonia groups when admitted with equally elevated FBG levels (all *p* < 0.05). Based on these observations, we hypothesise that SARS-CoV-2 infection may damage the host regulation of FBG. SARS-CoV-2 needs to bind to ACE2 to infect cells, and this binding can interfere with glucose metabolism and damage relevant metabolic organs [13,23,42]. A recent report suggests that pancreatic endocrine cells that highly express ACE2 can be readily targeted by SARS-CoV-2 [43,44]. The spike protein of SARS-CoV-2 binds to ACE2, and with the help of host transmembrane serine protease 2 (TMPRSS2), both proteins—being present in the human pancreas—may facilitate the entry of SARS-CoV-2 into pancreatic cells [44,45]. Hence, the damage to the endocrine pancreas caused by SARS-CoV-2 infection may increase the risk of new-onset diabetes in COVID-19 patients. Furthermore, increased inflammation and dysregulated immune system responses caused by SARS-CoV-2 infection could also disrupt endocrine signalling and result in hyperglycaemia [46,47]. People in self-isolation or lockdown may have reduced physical activity and poor diet, leading to weight gain, increased insulin resistance and, ultimately, increased risk of hyperglycaemia.

Our study has several limitations. Firstly, the data collection dates and venues of the COVID-19 and bacterial pneumonia patients differed, leading to potential selection bias in the study participants, despite both hospitals in different locations being managed and operated under the same administration. This study was limited to areas within China. Future studies with pneumonia patients collected from multiple hospitals would be useful to confirm our findings. Secondly, pneumonia patients with community-acquired bacterial infections may choose to terminate treatment and discharge from the hospital when they receive a notification of imminent death. As a result, complete death data cannot always be obtained in the hospital setting. This may underestimate the death rate and cause bias in our analysis of the association between FBG levels and mortality rates in pneumonia patients with bacterial and concurrent infections.

## 5. Conclusions

Our study concludes that an elevated FBG level at the time of hospital admission contributes to subsequent clinical complications among pneumonia patients, regardless of the underlying aetiology. However, poor clinical prognosis is more common in COVID-19 patients with elevated FBG levels than in pneumonia patients with sole bacterial infection or concurrent infections. Furthermore, COVID-19 patients tend to sustain high FBG levels throughout their hospitalisation, and even at discharge, compared to other pneumonia groups.

## Figures and Tables

**Figure 1 pathogens-11-00902-f001:**
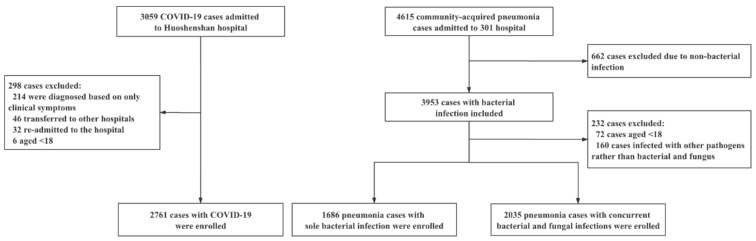
Flowchart describing the selection process of participants (COVID-19, pneumonia patients with sole bacterial infection, or pneumonia patients with concurrent bacterial and fungal infections).

**Figure 2 pathogens-11-00902-f002:**
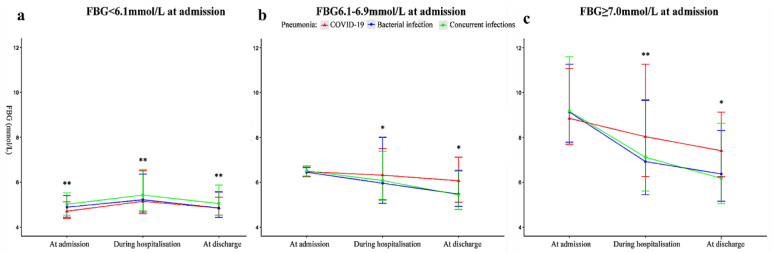
Comparison of changes in fasting blood glucose levels during hospitalisation and at discharge among the three pneumonia patient groups. The patients from the 3 pneumonia groups with their basal FBG levels of <6.1 mmol/L (**a:** left panel), 6.1–6.9 mmol/L (**b:** middle panel), and ≥7.0 mmol/L (**c:** right panel) at admission were compared for the changes in their FBG levels during hospitalisation and at discharge. FBG levels are shown as the median and interquartile range (IQR) at the 3 specified time points; * *p* < 0.05 vs. COVID-19 group; ** *p* < 0.001 vs. COVID-19 group. The participants with missing data for FBG levels at admission within 72 h were not included for analysis.

**Figure 3 pathogens-11-00902-f003:**
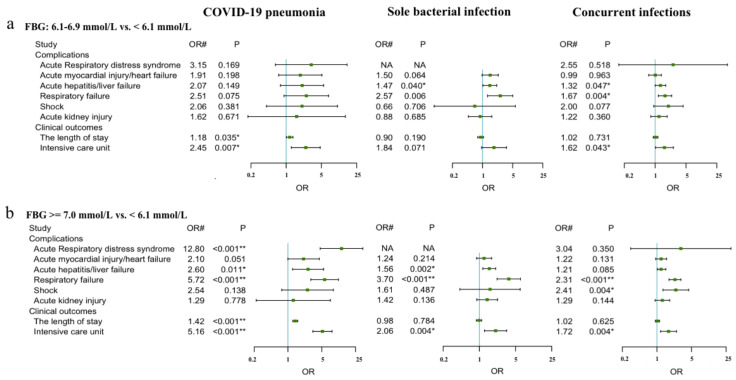
Adjusted odds ratios of complications and hospital outcomes in patients with pneumonia by FBG levels at admission: The odds ratios for FBG levels at admission of 6.1–6.9 mmol/L versus < 6.1 mmol/L (**a**) and ≥7 mmol/L versus <6.1 mmol/L (**b**) for complications and clinical outcomes in the three pneumonia groups are shown. NA: separate variable; * *p* < 0.05 vs. participants with FBG < 6.1 mmol/L at admission; ** *p* < 0.001 vs. participants with FBG < 6.1 mmol/L at admission; OR^#^: standard OR. Parameters adjusted include age, sex, and pre-existing comorbidities (i.e., chronic obstructive pulmonary disease, chronic kidney disease, cerebrovascular disease, coronary heart disease, cancer, chronic bronchitis, hepatitis, and hypertension).

**Figure 4 pathogens-11-00902-f004:**
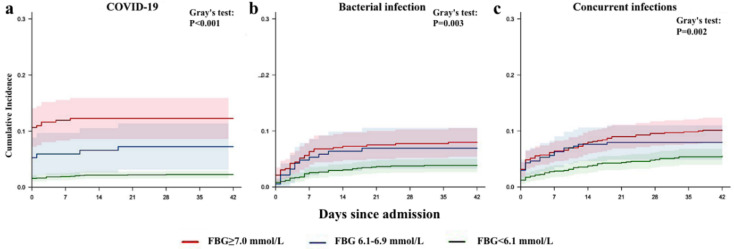
Cumulative incidence of ICU admission events during hospitalisation, stratified by patients with COVID-19 (**a**), sole bacterial infection (**b**), or concurrent bacterial and fungal infections (**c**) at specified FBG levels at admission (the participants with missing data for FBG levels on admission within 72 h were not included for analysis).

**Table 1 pathogens-11-00902-t001:** Basic demographic characteristics, signs and symptoms, pre-existing comorbidities, laboratory findings, and clinical outcomes of the three patient groups.

Variable	COVID-19	Pneumonia Patients with Bacterial Infections	Pneumonia Patients with Concurrent Infections	*p*-Value
	(*n* = 2761)	(*n* = 1686)	(*n* = 2035)	
**Demographic characteristics at admission**				
Age (years)—median (IQR)	60 (50, 68)	59 (47, 69)	65 (52, 79)	<0.001 **
<45—no. (%)	470 (17.0)	348 (20.6)	306 (15.0)	<0.001 **
45–59—no. (%)	865 (31.3)	520 (30.8)	461 (22.7)	
60–74—no. (%)	1133 (41.0)	529 (31.4)	575 (28.3)	
>74—no. (%)	293 (10.6)	289 (17.1)	693 (34.01)	
Male gender—no. (%)	1401 (50.7)	1170 (69.5)	1396 (68.7)	<0.001 **
Respiratory rate > 20 min—no. (%)	822 (29.9)	149 (9.6)	331 (17.6)	<0.001 **
Pulse rate > 100 per min—no. (%)	420 (15.3)	202 (12.9)	361 (19.1)	<0.001 **
Systolic blood pressure ≥140 mmHg—no. (%)	645 (25.4)	451 (29.0)	513 (27.3)	0.041 *
Diastolic blood pressure ≥90 mmHg—no. (%)	562 (22.1)	240 (15.4)	223 (11.9)	<0.001 **
**Signs and symptoms—no. (%)**				
Cough	1555 (56.6)	451 (26.8)	715 (35.1)	<0.001 **
Fatigue	1097 (39.7)	49 (2.9)	69 (3.4)	<0.001 **
Diarrhoea	79 (2.9)	12 (0.7)	24 (1.2)	<0.001 **
Chest tightness	331 (12.0)	134 (8.0)	178 (8.8)	<0.001 **
Shortness of breath	706 (25.6)	131 (7.8)	189 (9.3)	<0.001 **
**Pre-existing comorbidities—no. (%)**				
Coronary heart disease	160 (5.8)	292 (17.3)	492 (24.2)	<0.001 **
Cancer	47 (1.7)	549 (32.6)	539 (26.5)	<0.001 **
Chronic bronchitis	55 (2.0)	197 (11.7)	308 (15.1)	<0.001 **
Cerebrovascular disease	95 (3.4)	211 (12.5)	471 (23.1)	<0.001 **
Chronic kidney disease	64 (2.3)	391 (23.2)	551 (27.1)	<0.001 **
Chronic obstructive pulmonary disease	24 (0.9)	78 (4.6)	169 (8.3)	<0.001 **
Diabetes	375 (13.6)	333 (19.8)	467 (23.0)	0.020 *
Hepatitis	37 (1.3)	47 (2.8)	61 (3.0)	<0.001 **
Hypertension	810 (29.3)	571 (33.9)	837 (41.1)	<0.001 **
**Laboratory findings—median (IQR)**				
C-reactive protein (mg/L)	2.2 (0.8,8.5)	18.3 (3.5,70.0)	32.0 (8.6,87.8)	<0.001 **
D-dimer (mg/L)	0.4 (0.2,0.8)	0.9 (0.4,2.2)	1.6 (0.8,3.3)	<0.001 **
White blood cell count (10^9^/L)	5.7 (4.7,7.0)	7.1 (5.4,9.7)	8.2 (5.7,11.6)	<0.001 **
Lymphocyte ratio (%)	27.0 (20.3,32.9)	16.3 (7.2,27.4)	10.2 (4.0,18.9)	<0.001 **
Neutrophils ratio (%)	62.1 (55.7,69.5)	67.7(53.1,79.5)	74.5 (54.0,84.7)	<0.001 **
Monocyte ratio (%)	7.6 (6.2,9.0)	6.0 (3.8,7.8)	5.1 (2.1,7.3)	<0.001 **
Lactate dehydrogenase (IU/L)	176.5 (151.1,215.1)	183.2 (148.2,242.3)	210.7 (162.8,294.1)	<0.001 **
Thrombinogen time (s)	12.8 (12.2,13.5)	15.8 (15.0,16.6)	15.7 (14.9,16.7)	<0.001 **
Total bilirubin (μmol/L)	9.5 (7.3,12.3)	10.0 (7.1,14.5)	10.0 (7.1,15.2)	<0.001 **
Direct bilirubin (μmol/L)	3.3 (2.5,4.5)	3.1 (2.1,5.0)	3.5 (2.3,5.7)	<0.001 **
Albumin (g/L)	38.2 (35.3,40.5)	53.4 (48.0,57.9)	50.9 (46.1,56.0)	<0.001 **
Alkaline phosphatase (IU/L)	69.7 (58.3,84.7)	68.9 (54.8,87.9)	71.1 (55.6,91.2)	0.145
Fibrinogen (g/L)	3.0 (2.6,3.4)	4.4 (3.4,5.8)	4.7 (3.6,5.9)	<0.001 **
Creatinine (μmol/L)	64.1 (54.7,75.3)	73.5 (60.1,94.2)	72.1 (55.1,97.4)	<0.001 **
Creatine kinase (U/L)	50.6 (36.4,72.5)	62.9 (38.2,113.1)	53.4 (29.4,107.2)	<0.001 **
Creatine kinase-MB (IU/L)	8.6 (7.0,11.0)	2.1 (0.9,11.3)	1.8 (1.0,4.4)	<0.001 **
Fasting blood glucose (mmol/L)	4.9 (4.5,5.7)	5.5 (4.7,7.2)	6.1 (5.0,8.0)	<0.001 **
Cystatin C (mg/L)	0.9 (0.8,1.1)	1.1 (0.9,1.8)	1.2 (0.9,1.7)	<0.001 **
Platelets count (10^9^/L)	222.0 (181.0,272.0)	212.0 (161.0,274.0)	199.0 (141.0,267.8)	<0.001 **
**Complications—no. (%)**				
Acute respiratory distress syndrome	21 (0.8)	3 (0.2)	5 (0.3)	0.005 *
Acute myocardial injury/failure	45 (1.6)	494 (29.3)	834 (41.0)	<0.001 **
Acute hepatitis/liver failure	44 (1.6)	386 (22.9)	655 (32.2)	<0.001 **
Respiratory failure	46 (1.7)	86 (5.1)	331 (16.3)	<0.001 **
Shock	15 (0.5)	11 (0.7)	61 (3.0)	<0.001 **
Acute kidney injury	8 (0.3)	401 (23.8)	622 (30.6)	<0.001 **
**Clinical outcomes**				
Length of stay (days)—median (IQR)	14.0 (9.0,20.0)	14.0 (9.0,20.0)	20.0 (13.0,29.0)	<0.001 **
Intensive care unit—no. (%)	108 (3.9)	90 (5.3)	159 (7.8)	<0.001 **

* *p*-Value is between 0.05 and 0.001; ** *p*-value < 0.001.

## Data Availability

The data presented in this study are available on request from the corresponding author. The data are not publicly available due to restrictions apply to the availability of these data, which were used under license for the current study.

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
