# Peer review of "Elevated Fasting Blood Glucose Levels Are Associated with Worse Clinical Outcomes in COVID-19 Patients Than in Pneumonia Patients with Bacterial Infections"

_pathogens, 2022, doi:10.3390/pathogens11080902_

Round 1

Reviewer 1 Report

Dear colleagues!

Coronavirus remains an urgent public health problem and the issues of predicting complications based on laboratory tests are still open.  I would like to thank the authors for an interesting study.

On line 75 you write about diabetes, but it is fair to say that other pathologies increase the risk of mortality. This should be mentioned and somewhat expanded on line 68.

The Materials and Methods section is written clearly, however, ethics issues and permissions of the relevant committee for the study should be clarified.

In the results figures 2, 3 and 4 are poorly readable and contain too many details - it is advisable to change them to make them more informative.

In references, links 24 and 25 are the same, unfinished links in 39 and 43.

Author Response

Coronavirus remains an urgent public health problem and the issues of predicting complications based on laboratory tests are still open.  I would like to thank the authors for an interesting study.

  1. On line 75 you write about diabetes, but it is fair to say that other pathologies increase the risk of mortality. This should be mentioned and somewhat expanded on line 68.

Response: Thank you for your suggestion. We have now revised the relevant text of paragraph 3 in introduction that reads: “Apart from the complications mentioned above, previous studies have also shown that increased C-reactive protein, diabetes, hyperglycaemia and other clinical indicators are associated with increased risk of mortality in pneumonia patients[3][8], among which diabetes or hyperglycaemia is also a strong risk factor for mortality in community-acquired bacterial pneumonia (CAP-bacteria)[10].”

  1. The Materials and Methods section is written clearly, however, ethics issues and permissions of the relevant committee for the study should be clarified.

Response: The ethics approval and the details of the ethics committee are provided in the “ Ethics approval and consent to participate” section before the Reference section.

  1. In the results figures 2, 3 and 4 are poorly readable and contain too many details - it is advisable to change them to make them more informative.

Response: We have increased the font size of the text in the figures to make them more readable.

  1. In references, links 24 and 25 are the same, unfinished links in 39 and 43.

Response: We have deleted the duplicate reference 25 and revised the reference 39 and 43, which reads now:

[39] Mozafari N, Azadi S, Mehdi-Alamdarlou S, Ashrafi H, Azadi A. Inflammation: A bridge between diabetes and COVID-19, and possible management with sitagliptin. Medical hypotheses 2020; 143: 11011.

[43] Millet JK, Whittaker GR. Host cell proteases: Critical determinants of coronavirus tropism and pathogenesis. Virus Research 2015; 202:120-134.”

Reviewer 2 Report

In their manuscript, ‘Elevated fasting blood glucose level predicts worse clinical outcomes in COVID-19 patients than pneumonia patients with bacterial infection’, Wang et al compared the association between fasting blood sugar and clinical outcomes in COVID-19 patients compared to pneumonia of other etiology patients. The author conclude that FBG is significantly associated with worse outcomes in COVID19 patients and the outcomes are more severe than among patients with other causes for pneumonia. The manuscript is well-written however, I am concerned about selection bias. The following issues must be resolved.

Major issues

1-      Selection bias is a major concern, beyond what the authors mention in their limitations section. The non-COVID arms need to have been verified to be COVID free and this is not mentioned in the study. If this verification was not done, then the study design would be significantly flawed to provide reliable conclusions.

2-      The exclusion criteria for pneumonia patients is not clear, which infections, if not bacterial or fungal, were excluded?

3-      Figure 3: Reporting the ORs as non-logarithmic but then using log in the graph is confusing, keep the non-logarithmic format for both cases.

4-      The title must be changed to remove the word ‘predicts’. To determine a predictor a stronger study design than a retrospective cohort would be needed. The correct term to use is ‘associated’.

Minor issues

Line 55: replace ‘like diabetes and AIDS’ with ‘such as diabetes and AIDS patients’

Line 111: replace ‘have the same level’ with ‘are similar’

Line 144: How were the interactions terms generated

Line 156: Rephrase the sentence relevant to age ‘and proportion of male’ for clarity

Author Response

In their manuscript, ‘Elevated fasting blood glucose level predicts worse clinical outcomes in COVID-19 patients than pneumonia patients with bacterial infection’, Wang et al compared the association between fasting blood sugar and clinical outcomes in COVID-19 patients compared to pneumonia of other etiology patients. The author conclude that FBG is significantly associated with worse outcomes in COVID19 patients and the outcomes are more severe than among patients with other causes for pneumonia. The manuscript is well-written however, I am concerned about selection bias. The following issues must be resolved.

Major issues

  1. Selection bias is a major concern, beyond what the authors mention in their limitations section. The non-COVID arms need to have been verified to be COVID free and this is not mentioned in the study. If this verification was not done, then the study design would be significantly flawed to provide reliable conclusions.

Response: The 4,615 non-COVID pneumonia patients were admitted to the Hospital in Beijing from 6th April 2012 - 24th April 2020 (8 years). Most of the subjects were from the prior-Covid-19 era, with about 6 subjects admitted to the Hospital in Beijing in January-April 2020. These patients admitted to this hospital in 2020 were all confirmed to be non-Covid-19, as all the Covid-19 patients are transferred to Covid-19 dedicated hospitals in Beijing. Therefore, all the 4,615 pneumonia patients in this category were from prior-Covid-19 era or confirmed to be non-Covid-19.

We have now revised the paragraph 1 in methods that “We established a retrospective observational cohort study based on 3,059 COVID-19 patients admitted to the Huoshenshan hospital in Wuhan, an emergency hospital rapidly constructed and dedicated to COVID-19 patients, from 4th February - 15th April 2020 and 4,615 pneumonia patients admitted to the Chinese PLA General Hospital in Beijing from 6th April 2012 - 24th April 2020. Majority of the pneumonia patients were from the prior-COVID-19 era and those admitted to the hospital after COVID-19 started were confirmed to be non-COVID-19, whereas the COVID-19 patients were transferred to other specific hospitals for treating COVID-19 patients. The two hospitals and their laboratories were under the same administration and then their workers, laboratory tests and treatments have the same level. We categorised the participants into three groups, including COVID-19 patients, pneumonia patients with sole bacterial infection, and those with concurrent bacterial and fungal infections. Details of inclusion and exclusion criteria and hospital discharge criteria were provided in Table S1 of supplemental materials.”

  1. The exclusion criteria for pneumonia patients is not clear, which infections, if not bacterial or fungal, were excluded?

Response: We appreciate the positive comments. Since the main topic in our manuscript assess the associations between FBG levels and the clinical severities between Covid-19 patients and patients infected by bacteria. Pneumonia patients with concurrent bacteria and fungal is most common. Therefore, the pneumonia patients with concurrent infections mainly include patients with bacteria and fungus. In patients selection, we only include the patients infected by bacteria or concurrent bacteria and fungus.

We have now revised the exclusion criteria in supplemental materials that “ Pneumonia patients with sole bacterial infection or concurrent bacterial and fungal infections: exclusion criteria: 1.patients younger than 18 years old; 2. patients with pneumonia with infection by pathogen(s) other than bacterium or bacterium and fungus, such as virus, mycoplasma, chlamydia, parasite and abiotic etiology.”

  1. Figure 3: Reporting the ORs as non-logarithmic but then using log in the graph is confusing, keep the non-logarithmic format for both cases.

Response: We agree and have changed ORs back to non-logarithmic format.

  1. The title must be changed to remove the word ‘predicts’. To determine a predictor a stronger study design than a retrospective cohort would be needed. The correct term to use is ‘associated’.

Response: We agree with the reviewer and have revised the title that reads now as “Elevated fasting blood glucose level is associated with worse clinical outcomes in COVID-19 patients than pneumonia patients with bacterial infection.”

Minor issues

  1. Line 55: replace ‘like diabetes and AIDS’ with ‘such as diabetes and AIDS patients’

Response: done.

  1. Line 111: replace ‘have the same level’ with ‘are similar’

Response: The two hospitals and their laboratories are under the same administration and then their workers, laboratory tests and treatments are similar.

  1. Line 144: How were the interactions terms generated

Response: We have now revised the paragraph 4 in methods that reads now as below:

The interaction was used to assess the impact of interactions among FBG levels and three pneumonia groups on clinical outcomes, as previously described [27].

[27] Institute of Medicine Committee on Assessing Interactions Among Social, Behaviroral,  

and Genetic Factors in Health; Hernandez LM, Blazer DG, editors. Genes, Behavior, and the Social Environment: Moving Beyond the Nature/Nurture Debate. Washington (DC): National Academies Press (US); Study Design and Analysis for Assessment of Interactions. August 2006.

  1. Line 156: Rephrase the sentence relevant to age ‘and proportion of male’ for clarity

Response: We have now revised the paragraph 1 in results that “Pneumonia patients with concurrent infections were consistently older than patients with sole bacterial infection (median 65 [52-79] years versus 59 [47-69] years) and COVID-19 patients (median 65 [52-79] years versus 60 [50-68] years). There was a greater male bias in Pneumonia patients with sole bacterial infection than patients with concurrent infections and COVID-19 patients (69.48% vs 68.67% and 50.74%, respectively). ”

Round 2

Reviewer 2 Report

A final revision to correct some minor language/grammar issues would help

Author Response

We appreciate the positive comments.

We have modified the prior version of the paper based on the comments. We reviewed the manuscript and corrected some minor language/grammar mistakes. All these changes made to the manuscript are marked up with track changes as required and marked in red.

This manuscript is a resubmission of an earlier submission. The following is a list of the peer review reports and author responses from that submission.